# Prolonging the integrated stress response enhances CNS remyelination in an inflammatory environment

Yanan Chen[1], Rejani B Kunjamma[1], Molly Weiner[1], Jonah R Chan[2], Brian Popko[1]*

[1]Department of Neurology, Division of Multiple Sclerosis and Neuroimmunology, Northwestern University Feinberg School of Medicine, Chicago, United States; [2]Weill Institute for Neuroscience, Department of Neurology, University of California, San Francisco, San Francisco, United States

**Abstract** The inflammatory environment of demyelinated lesions in multiple sclerosis (MS) patients contributes to remyelination failure. Inflammation activates a cytoprotective pathway, the integrated stress response (ISR), but it remains unclear whether enhancing the ISR can improve remyelination in an inflammatory environment. To examine this possibility, the remyelination stage of experimental autoimmune encephalomyelitis (EAE), as well as a mouse model that incorporates cuprizone-induced demyelination along with CNS delivery of the proinflammatory cytokine IFN-γ were used here. We demonstrate that either genetic or pharmacological ISR enhancement significantly increased the number of remyelinating oligodendrocytes and remyelinated axons in the inflammatory lesions. Moreover, the combined treatment of the ISR modulator Sephin1 with the oligodendrocyte differentiation enhancing reagent bazedoxifene increased myelin thickness of remyelinated axons to pre-lesion levels. Taken together, our findings indicate that prolonging the ISR protects remyelinating oligodendrocytes and promotes remyelination in the presence of inflammation, suggesting that ISR enhancement may provide reparative benefit to MS patients.

**\*For correspondence:**
brian.popko@northwestern.edu

## Introduction

Multiple sclerosis (MS) is an autoimmune inflammatory disorder characterized by focal demyelinated lesions in the CNS (*Frohman et al., 2006*; *Yadav et al., 2015*; *Reich et al., 2018*). Although current immunomodulatory therapies can reduce the frequency and severity of relapses, they have demonstrated limited impact on the progression of disease (*Dargahi et al., 2017*; *Hauser and Cree, 2020*). Complementary strategies are therefore urgently needed to protect oligodendrocytes and promote repair of the CNS in order to slow or even stop the progression of MS (*Hart and Bainbridge, 2016*; *Rodgers et al., 2013*; *Way and Popko, 2016*).

Remyelination is the process of restoring demyelinated nerve fibers with new myelin, which involves the generation of new mature oligodendrocytes from oligodendrocyte precursor cells (OPCs) (*Franklin and Ffrench-Constant, 2008a*). Failure of myelin repair during relapsing-remitting MS leads to chronically demyelinated axons, which is thought to contribute to axonal degeneration and disease progression (*Franklin and Kotter, 2008b*; *Fancy et al., 2010*; *Chari, 2007*). The inflammatory environment in MS lesions is considered a major contributor to impaired remyelination (*Starost et al., 2020*). Strategies to enhance myelin regeneration could preserve axonal integrity and increase clinical function of patients. A number of small molecules have recently been described to promote OPC maturation and/or enhance remyelination, but most of these molecules were identified in in vitro screens under ideal conditions and tested in animal models of non-inflammatory demyelination (*Deshmukh et al., 2013*; *Mei et al., 2014*; *Rankin et al., 2019*; *Najm et al., 2015*). Given that remyelination in MS occurs in lesions with ongoing inflammation, a remyelination model

with an inflammatory environment would provide a better indication of the potential of remyelinating-promoting compounds for MS treatment.

The integrated stress response (ISR) plays a key role in the response of oligodendrocytes to CNS inflammation (*Chen et al., 2019*; *Lin et al., 2005*; *Way et al., 2015*). The ISR is a cytoprotective pathway that is activated by various cytotoxic insults, including inflammation (*Costa-Mattioli and Walter, 2020*). The ISR is initiated by phosphorylation of eukaryotic translation initiation factor two alpha (p-eIF2$\alpha$), by one of four known stress-sensing kinases, to diminish global protein translation and selectively allow for the synthesis of protective proteins. p-eIF2$\alpha$ is dephosphorylated by the protein phosphatase 1 (PP1)- DNA-damage inducible 34 (GADD34) complex to terminate the ISR upon stress relief (*Zhang and Kaufman, 2008*). Sephin1, a recently-identified small molecule, disrupts PP1-GADD34 phosphatase activity and prolongs elevated p-eIF2$\alpha$ levels, thereby enhancing the ISR protective response (*Das et al., 2015*; *Carrara et al., 2017*). We previously showed that GADD34 deficiency or Sephin1 treatment protects matures oligodendrocytes, diminishes demyelination, and delays clinical symptoms in a mouse model of MS, experimental autoimmune encephalomyelitis (EAE) (*Chen et al., 2019*). It is not known whether a protracted ISR would protect newly generated remyelinating oligodendrocytes and enhance remyelination in an inflammatory environment.

Here, we investigated the potential impact of genetic or pharmacological ISR enhancement on remyelinating oligodendrocytes and the remyelination process in the presence of inflammation. In these studies, we used the EAE model, as well as the cuprizone-induced demyelination model on GFAP-tTA;TRE-IFN-$\gamma$ double-transgenic mice, which ectopically express interferon gamma (IFN-$\gamma$) in the CNS in a regulated manner (*Lin et al., 2004*; *Lin et al., 2005*). IFN-$\gamma$ is a pleotropic T-cell-specific cytokine that stimulates key inflammatory aspects of MS (*Lin et al., 2007*; *Lees and Cross, 2007*). Using these inflammatory demyelination models, we show that prolonging the ISR protects remyelinating oligodendrocytes and increases the level of remyelination. Moreover, we explored the effects of combining Sephin1 treatment with bazedoxifene (BZA) (*Rankin et al., 2019*) on remyelination. BZA has been shown to enhance OPC differentiation and CNS remyelination in response to focal demyelination (*Rankin et al., 2019*). We demonstrate that BZA increases the number of remyelinating oligodendrocytes and enhances remyelination in the presence of inflammation. When combined with Sephin1, BZA further facilitates the remyelinating process.

## Results

### Sephin1 treatment enhances remyelination in late-stage EAE

To determine whether pharmacological prolongation of the ISR can enhance remyelination after inflammatory demyelination, we first examined C57BL/6J mice in the late stage of EAE. Sephin1 (8 mg/kg) or vehicle treatment was initiated in each EAE mouse on the day it reached the peak of disease (clinical score = 3) and was continued to the late stage of EAE, which resulted in diminished EAE disease severity in the final week of Sephin1 treatment (*Figure 1A*). Spinal cord axons were examined under electron microscope (EM). We sorted remyelinated axons by examining g-ratios (axon diameter / total fiber diameter) in the lumbar spinal cord white matter of EAE mice after vehicle or Sephin1 treatment. The presence of axons with thinner myelin sheaths and a higher g-ratio is considered the hallmark of remyelination (*Duncan et al., 2017*). A recent study indicated that axons in the spinal cord with a g-ratio greater than 0.8 are likely remyelinated axons in the EAE model (*Mei et al., 2016*). Our data demonstrated that the density of remyelinated axons (g > 0.8) was significantly higher in EAE mice treated with Sephin1 than those treated with vehicle ($p < 0.05$, $\eta^2 = 0.81$) (*Figure 1B,C*), although no difference was detected in the total number of myelinated axons (*Figure 1D*). The data suggest that Sephin1 promotes remyelination in the neuroinflammatory environment of EAE, presumably by protecting remyelinating oligodendrocytes against inflammatory stress.

### GADD34 deficiency protects remyelinating oligodendrocytes and enhances remyelination in the presence of IFN-$\gamma$

Our finding of increased remyelination in the Sephin1 treated late stage EAE mice raised the possibility that prolonging the ISR might provide protection against inflammation to remyelinating

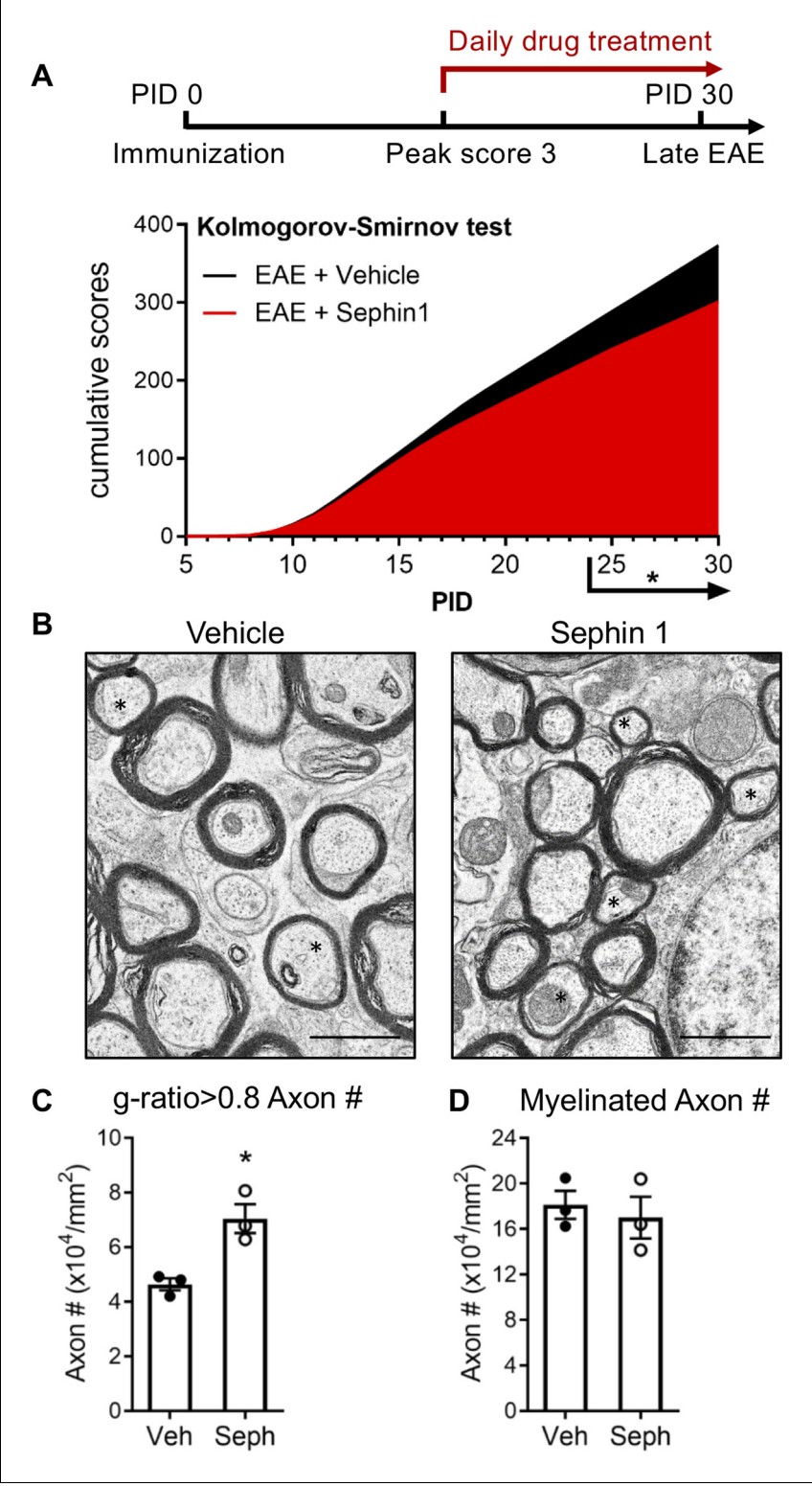

**Figure 1.** Sephin1 treatment enhances remyelination after inflammatory demyelination. (**A**) Cumulative clinical scores of C57BL/6J female mice immunized with $MOG_{35-55}$/CFA to induce chronic EAE, treated with vehicle ($n = 7$) and 8 mg/kg Sephin1 ($n = 7$) from the peak disease. *$p<0.05$. Significance based on Kolmogorov-Smirnov test. (**B**) Representative EM images of axons in the spinal cord white matter tracts of EAE mice treated with vehicle or Sephin1 at PID30. Remyelinated axons in the EAE spinal cord were identified by thinner myelin sheaths (*). Scale bar = 1 μm. (**C**) Density of myelinated axons with g-ratio <0.8 in the EAE spinal cord. (**D**) Density of

*Figure 1 continued on next page*

*Figure 1 continued*

total myelinated axons in the EAE spinal cord. Data are presented as the mean ± SEM (n = 3 mice/group). Over 300 axons were analyzed per mouse. *p<0.05. Significance based on unpaired *t*-test.

The online version of this article includes the following source data for figure 1:

**Source data 1.** Disease scores and axon measurement of late stage of EAE.

oligodendrocytes and thereby promote remyelination. To examine this possibility, we used a more quantitative demyelination/remyelination model (cuprizone model) for further exploration. We have previously shown that in response to IFN-γ, myelinating oligodendrocytes with a *Ppp1r15a (Gadd34)* mutation display increased levels of p-eIF2α, indicating a prolongation of the ISR, and increased oligodendrocyte survival (*Lin et al., 2008*). We hence mated GADD34 KO;GFAP-tTA mice with GADD34 KO;TRE-IFN-γ mice to generate GFAP-tTA;TRE-IFN-γ double-transgenic mice homozygous for the *Ppp1r15a* mutation (GADD34 KO) to prolong the ISR (*Lin et al., 2008*). GFAP-tTA;TRE-IFN-γ double-transgenic mice allow for ectopic release of IFN-γ in the CNS in a doxycycline (Dox)-dependent manner (*Lin et al., 2004*; *Lin et al., 2006*; *Lin et al., 2008*). Expression of the tetracycline-controlled transactivator (tTA) is driven by the astrocyte-specific transcriptional regulatory region of the *Gfap* gene. In the TRE-IFN-γ mice, the IFN-γ cDNA is transcriptionally controlled by the tetracycline response element (TRE) (*Figure 2A*). The expression of the IFN-γ transgene is repressed in the GFAP-tTA;TRE-IFN-γ mice by providing water treated with Dox from conception. At 6 weeks of age, transgenic mice were taken off Dox water to induce CNS expression of IFN-γ (IFN-γ+) and placed on a diet of 0.2% cuprizone chow (*Lin et al., 2006*; *Lin et al., 2004*). After 5 weeks of cuprizone exposure, mice were placed back on a normal diet for up to 3 weeks to allow remyelination to occur. Control mice received Dox water throughout the study to repress CNS expression of IFN-γ (IFN-γ-) (*Figure 2B*). Mice with the highest level of IFN-γ expression in the CNS were selected by isolating the cerebellum of each mouse and using real-time reverse transcription (RT)-PCR to determine IFN-γ expression levels. We found that at 5 weeks of cuprizone exposure (W5) and during remyelination

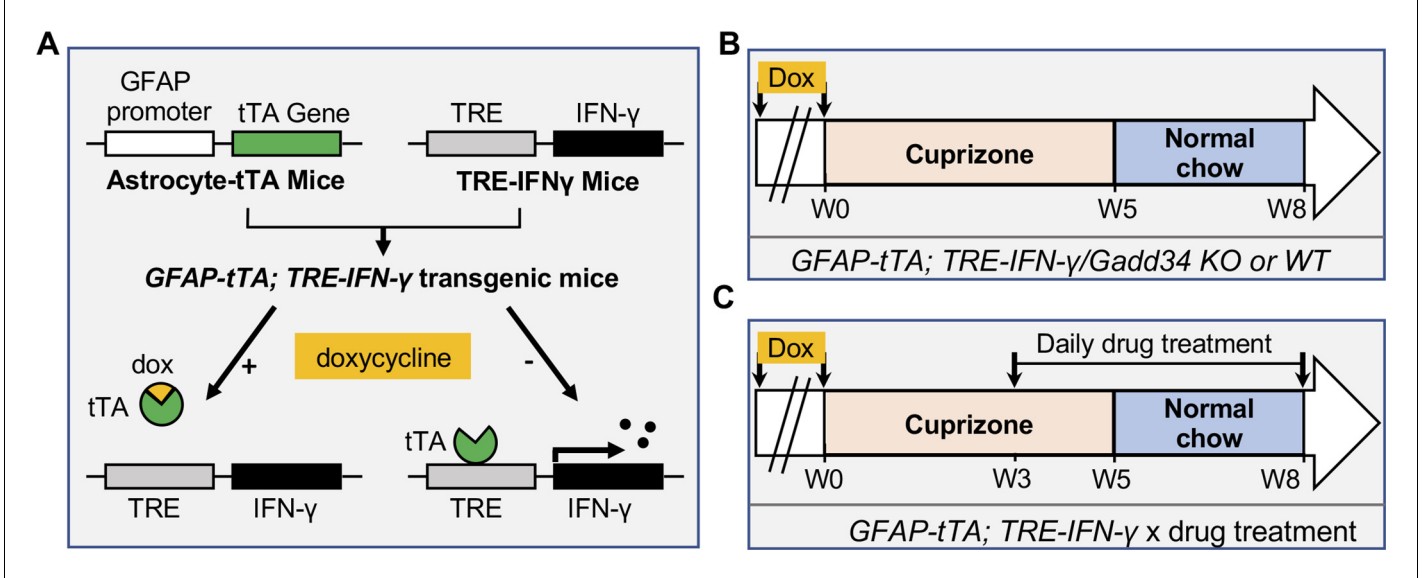

**Figure 2.** Schematics of GFAP-tTA;TRE-IFN-γ double-transgenic mouse model of cuprizone demyelination/remyelination. (**A**) GFAP-tTA mice are mated with TRE-IFN-γ mice to produce double-positive animals. When these mice are maintained on doxycycline (Dox), the expression of the IFN-γ is repressed. When they are released from Dox, IFN-γ is expressed in the CNS. (**B**) Cuprizone demyelination/remyelination model of GFAP-tTA;TRE-IFN-γ/GADD34 KO or WT. Dox is removed and cuprizone chow is added when mice are at 6-week-old (W0). After 5 weeks of cuprizone exposure (W5), mice were placed back on normal chow for up to 3 weeks to allow remyelination. (**C**) Cuprizone demyelination/remyelination model of GFAP-tTA;TRE-IFN-γ with designed treatment. Drug treatment is started at 3 weeks of cuprizone exposure (W3) and lasts to the end of remyelination (W8).

The online version of this article includes the following figure supplement(s) for figure 2:

**Figure supplement 1.** GFAP-tTA;TRE-IFN-γ mice express IFN-γ after release from doxycycline.

(W8), removal of Dox significantly increased the levels of IFN-γ in the cerebellum of mice (*Figure 2—figure supplement 1A*).

We next investigated whether the *Gadd34* mutation promotes oligodendrocyte survival in the presence of IFN-γ during demyelination/remyelination in the cuprizone model. It has been demonstrated that cuprizone-fed mice exhibit apoptotic death of oligodendrocytes and demyelination in the corpus callosum. Complete remyelination spontaneously occurs a few weeks after the cuprizone challenge is terminated (*Matsushima and Morell, 2001*). We found no significant difference in the number of cells labeled with the mature oligodendrocyte marker ASPA between GADD34 wildtype (WT) and GADD34 KO, GFAP-tTA;TRE-IFN-γ double-transgenic mice before cuprizone exposure (*Figure 3A,B*). After 5 weeks on cuprizone chow, both GADD34 WT and KO double-transgenic mice presented similarly with a significantly reduced number of ASPA+ mature oligodendrocytes in the presence of IFN-γ (IFN-γ+) (WT, p<0.001; KO, p<0.05) (*Figure 3A,B*). During the remyelination period, ASPA+ oligodendrocytes reappeared and reached approximately 800/mm$^2$ in both IFN-γ-repressed GADD34 WT and KO mice (IFN-γ-) after 3 weeks of normal chow. In the presence of IFN-γ (IFN-γ+), this number dropped in the lesions of WT double-transgenic mice to roughly 500/mm$^2$ of ASPA+ cells (p<0.05) (*Figure 3A,C*), which is consistent with our previous finding that IFN-γ expression in the CNS suppresses the repopulation of oligodendrocytes following cuprizone-induced oligodendrocyte toxicity (*Lin et al., 2006*). Compared to GADD34 WT, GFAP-tTA;TRE-IFN-γ mice with the *Gadd34* mutation had significantly increased numbers of ASPA+ oligodendrocytes during remyelination in the presence of IFN-γ (IFN-γ+/W8) (p<0.01) (*Figure 3A,C*). Myelin status in these mice was also evaluated with EM. After 5 weeks of cuprizone chow, a significant number of axons exhibited myelin sheath loss in control mice (IFN-γ-), compared to 0 weeks (WT, p<0.0001; KO, p<0.001) (*Figure 4A,C*). Similarly, both WT and KO double-transgenic mice presented with markedly reduced numbers of myelinated axons in the presence of IFN-γ (IFN-γ+) (WT: 25.2 ± 3.0%, p<0.0001; KO: 24.4 ± 2.7%, p<0.0001). Nevertheless, no difference in the percentage of myelinated axons was observed between IFN-γ+ and IFN-γ- groups (*Figure 4A,C*). At 3 weeks after cuprizone withdrawal, both IFN-γ-repressed WT and KO mice (IFN-γ-) showed spontaneous remyelination (WT: 42.7 ± 6.2%; KO: 42.6 ± 4.7%) (*Figure 4B,D*). Meanwhile, remyelination was significantly suppressed in the corpus callosum of IFN-γ-expressing WT mice (WT: 29.4 ± 2.9%) (p<0.05), but not in that of IFN-γ-expressing KO mice (KO: 49.1 ± 8.8%) (*Figure 4B,D*). Together, these data suggest that prolonging the ISR has no impact in the absence of IFN-γ, but it results in increased repopulation of oligodendrocytes and remyelination following demyelination in the presence of IFN-γ.

## Sephin1 treatment protects remyelinating oligodendrocytes and enhances remyelination in the presence of IFN-γ

We have previously demonstrated that Sephin1 treatment can prolong the ISR in primary oligodendrocytes exposed to IFN-γ, as shown by prolonged elevated eIF2α phosphorylation levels (*Chen et al., 2019*). In addition, Sephin1 protects mature oligodendrocytes against inflammation in the EAE model (*Chen et al., 2019*). Therefore, using Sephin1, we tested the ability of pharmacological enhancement of the ISR to protect remyelinating oligodendrocytes from inflammatory insults in the cuprizone model. Similar to the GADD34 mouse experiment, 6-week-old GFAP-tTA;TRE-IFN-γ double transgenic mice that had been on Dox water since conception were either kept on (IFN-γ-) or removed from Dox (IFN-γ+) and fed with a chow containing 0.2% cuprizone for 5 weeks. Mice were then placed back on a normal diet for 3 weeks. Sephin1 (8 mg/kg) or vehicle was administered daily to the mice by intraperitoneal injections beginning 3 weeks after the start of the cuprizone diet. This time point represents the peak of oligodendrocyte loss during cuprizone-mediated demyelination (*Matsushima and Morell, 2001*; *Figure 2C*).

We first examined IFN-γ expression in the double-transgenic mice, which showed that mice removed from Dox (IFN-γ+) expressed significantly higher levels of IFN-γ in the cerebellum than mice on Dox water (IFN-γ-) at 5 weeks and 8 weeks of cuprizone treatment in all groups (*Figure 2—figure supplement 1B*). Next, we investigated the number of mature oligodendrocytes in the corpus callosum by immunofluorescent staining of ASPA. 3 weeks of cuprizone exposure led to a significant decrease of the ASPA+ oligodendrocytes in both IFN-γ- (p<0.01) and IFN-γ+ (p<0.01) groups (*Figure 5A,B*). After 5 weeks of cuprizone chow, we did not detect further oligodendrocyte loss in the absence of IFN-γ. It has been shown that oligodendrocytes start regenerating after significant loss at 3 weeks, even in the presence of cuprizone (*Armstrong et al., 2006*). We observed a further

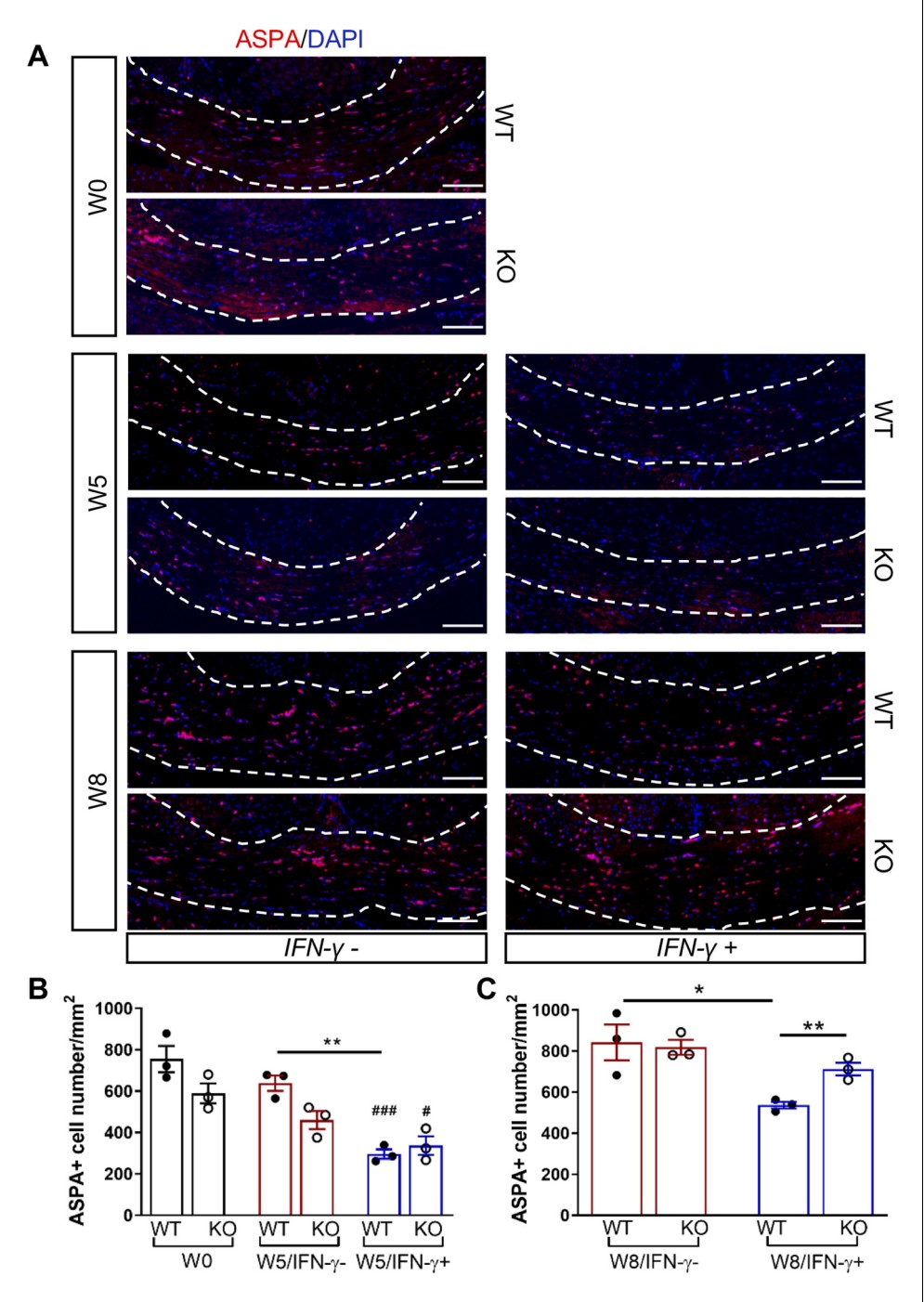

**Figure 3.** GADD34 deficiency protects remyelinating oligodendrocytes in the presence of IFN-γ. The corpora callosa of GFAP-tTA;TRE-IFN-γ/GADD34 KO or WT were taken at W0, W5, and W8. (**A**) Immunofluorescent staining for ASPA (a mature oligodendrocyte marker) and DAPI (nuclei). Scale bar = 100 μm. (**B**) Quantification of cells positive for ASPA in the corpus callosum areas at W0 and W5 in the absence (IFN-γ-) or presence of IFN-γ (IFN-γ+). Data are presented as the mean ± SEM (n = 3 mice/group). WT: eight males and one female; KO: five males and four females. **p<0.01, #p<0.05 (#vs W0/WT), ###p<0.001 (# vs W0/KO). Significance based on ANOVA. (**C**) Quantification of cells positive for ASPA in the corpus callosum areas at W8 in the absence or presence of IFN-γ. WT: two males and four females; KO: four males and two females. Data are presented as the mean ± SEM (n = 3 mice/group). *p<0.05, **p<0.01. Significance based on ANOVA.

The online version of this article includes the following source data for figure 3:

**Source data 1.** The number of ASAP+ cells in WT and KO.

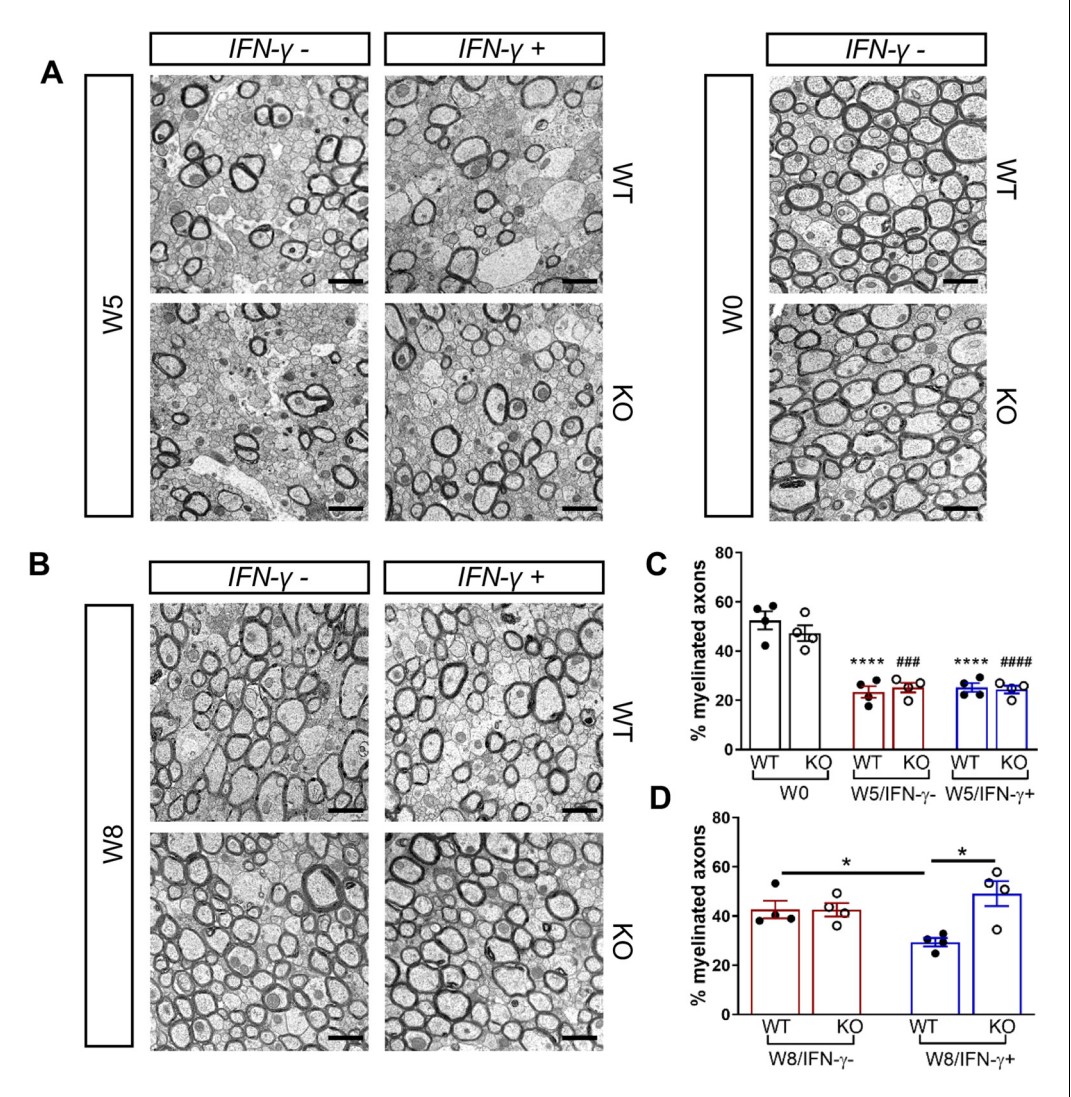

**Figure 4.** GADD34 deficiency enhances remyelination in the presence of IFN-γ. The corpora callosa of GFAP-tTA; TRE-IFN-γ/GADD34 KO or WT were harvested for EM processing. (**A**) Representative EM images of axons in the corpus callosum at W0 and W5. Scale bar = 1 μm. (**B**) Representative EM images of axons in the corpus callosum at W8. Scale bar = 1 μm. (**C**) Percentage of myelinated axons at W0 and W5. WT: nine males and three females; KO: six males and six females. ****p<0.0001 (* vs W0/WT); ###p<0.001, ####p<0.0001 ( #vs W0/KO). Significance based on ANOVA. (**D**) Percentage of myelinated axons at W8. WT: four males and four females; KO: five males and three females. Data are presented as the mean ± SEM (n = 4 mice/group). *p<0.05 Significance based on ANOVA.

The online version of this article includes the following source data for figure 4:

**Source data 1.** The number of myelinated axons in WT and KO.

reduction in the number of oligodendrocytes in the presence of IFN-γ (p<0.01), although we found no difference between vehicle and Sephin1 treatment (veh: 315 ± 44/mm²; Seph 394 ± 67/mm²) (*Figure 5A,B*). At 3 weeks after cuprizone withdrawal, the density of oligodendrocytes returned to the initial levels seen at week 0 (W0) in both vehicle- and Sephin1-treated mice in the absence of IFN-γ. Activation of IFN-γ expression by removing Dox (IFN-γ+) suppressed the repopulation of ASPA+ oligodendrocytes in the vehicle treated mice, but Sephin1 treatment significantly increased

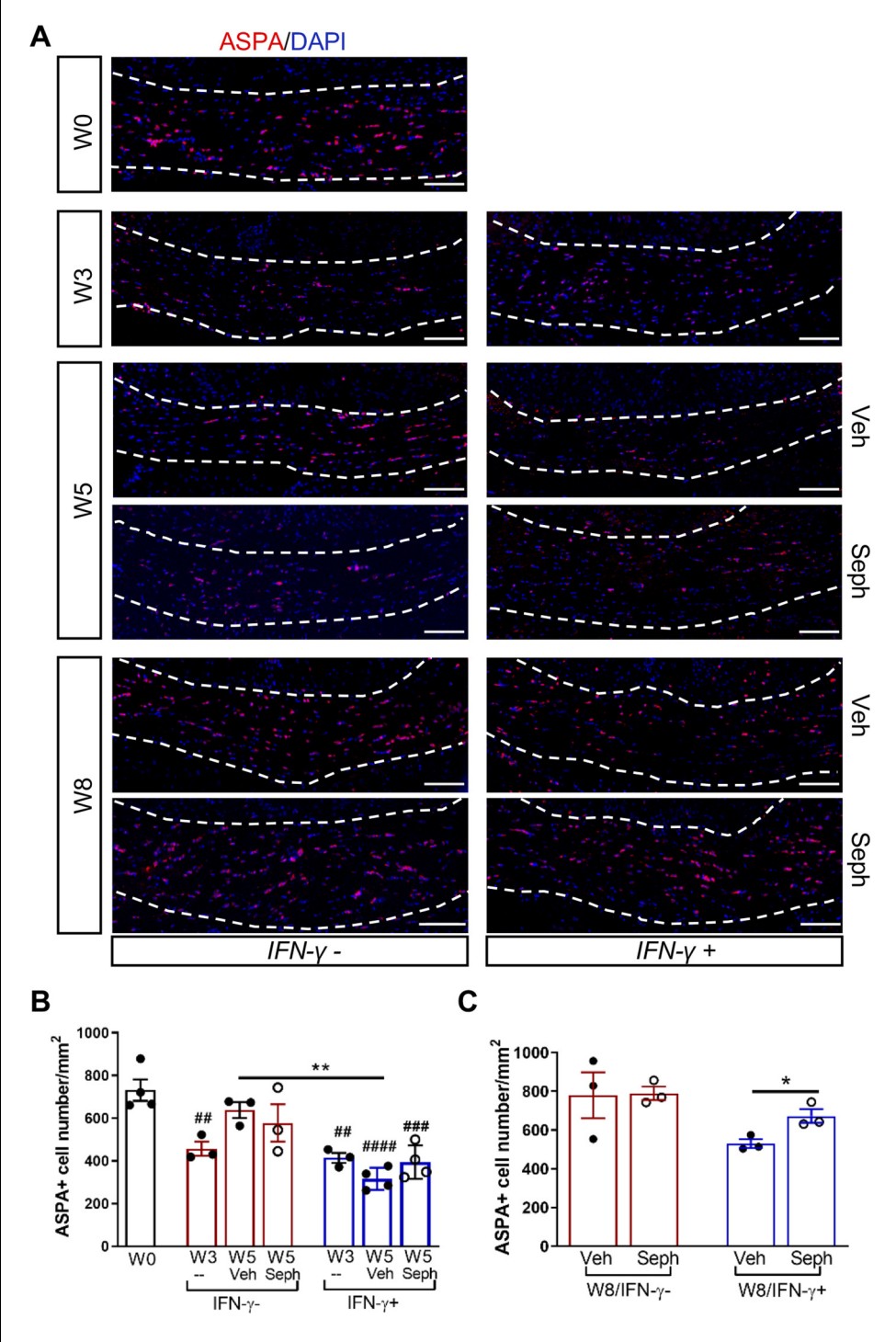

**Figure 5.** Sephin1 protects remyelinating oligodendrocytes in the presence of IFN-γ. The corpora callosa of GFAP-tTA;TRE-IFN-γ were taken at W0 and W3 prior to any treatment as well as after either vehicle or Sephin1 treatment at W5 and W8. (**A**) Immunofluorescent staining for ASPA (a mature oligodendrocyte marker) and DAPI (nuclei). Scale bar = 100 μm. (**B**) Quantification of cells positive for ASPA in the corpus callosum areas at W0, W3, and W5 in the absence (IFN-γ-) or presence of IFN-γ (IFN-γ+). Data are presented as the mean ± SEM (n = 3–4 mice/group). W0: four males; W3: five males and one female; W5 (Veh): four males and two females; W5 (Seph): four males and two females. **p<0.01, ##p<0.01, ###p<0.001, ####p<0.0001 (#vs W0). Significance based on ANOVA. (**C**) Quantification of cells positive for ASPA in the corpus callosum areas at W8 in the absence or

*Figure 5 continued on next page*

*Figure 5 continued*
presence of IFN-γ. Data are presented as the mean ± SEM (n = 3 mice/group). W8 (Veh): five males and one female; W8 (Seph): five males and one female. *p<0.05. Significance based on ANOVA.
The online version of this article includes the following source data for figure 5:

**Source data 1.** The number of ASAP+ cells in Veh and Seph groups.

the survival of these cells (veh: 531 ± 32/mm$^2$, Seph: 672 ± 52/mm$^2$, p<0.05, η$^2$ = 0.74) (*Figure 5A, C*). In addition, we observed an approximate 50% reduction in the percentage of myelinated axons after 5 weeks of cuprizone exposure in both IFN-γ- and IFN-γ+ double-transgenic mice (*p<0.0001*) (*Figure 6A,C*). Similar to the changes in oligodendrocyte numbers, our data demonstrated that during remyelination, CNS delivery of IFN-γ resulted in diminished numbers of remyelinated axons (p<0.05), while mice treated with Sephin1 exhibited a significantly higher percentage of remyelinated axons (veh: 30.3 ± 3.3%, Seph: 43.2 ± 4.6%, p<0.01, η$^2$ = 0.72) (*Figure 6B,D*). These data indicate that although the pharmacological enhancement of the ISR with Sephin1 has no impact in the absence of IFN-γ, it can provide protection to remyelinating oligodendrocytes and promote remyelination in the presence of IFN-γ.

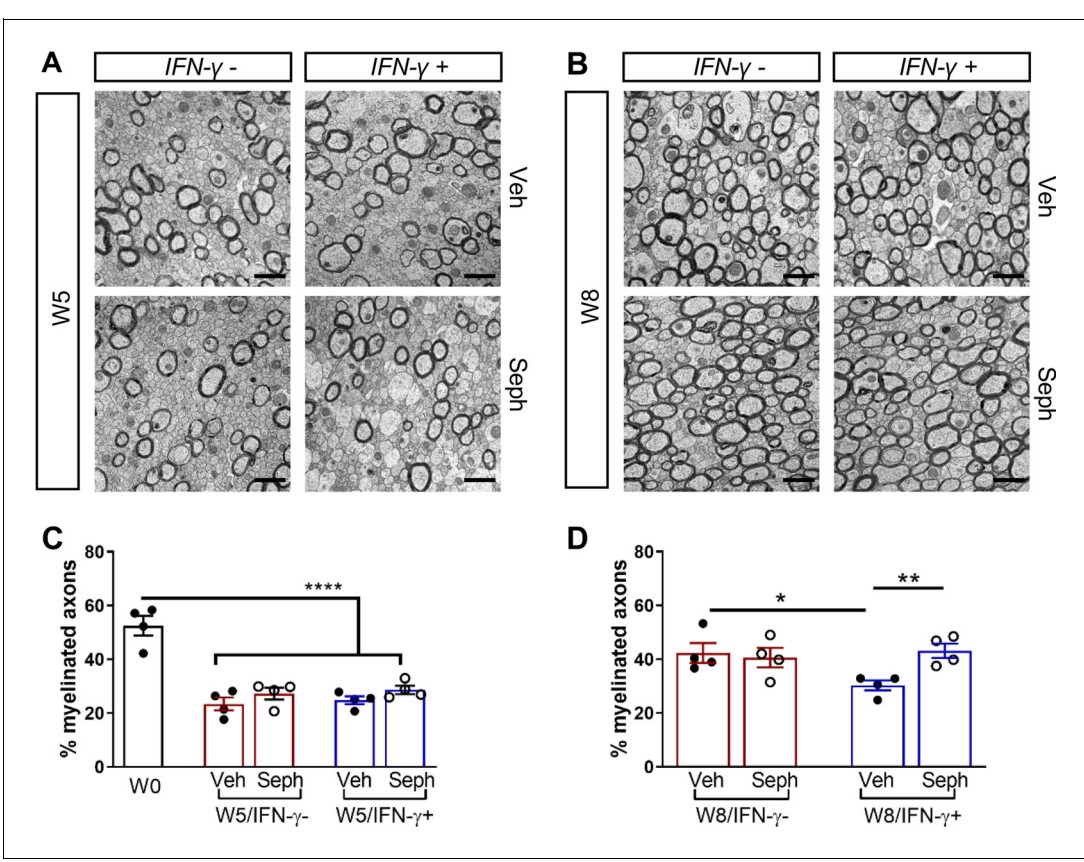

**Figure 6.** Sephin1 enhances remyelination in the presence of IFN-γ. The corpora callosa of GFAP-tTA;TRE-IFN-γ were harvested for EM processing. (**A**) Representative EM images of axons in the corpus callosum at W5. Scale bar = 1 μm. (**B**) Representative EM images of axons in the corpus callosum at W8. Scale bar = 1 μm. (**C**) Percentage of remyelinated axons at W0 and W5. W0: four males; W5 (Veh): five males and three females; W5 (Seph): five males and three females. (**D**) Percentage of remyelinated axons at W8. Data are presented as the mean ± SEM (n = 4 mice/group). W8 (Veh): six males and two females; W8 (Seph): six males and two females. *p<0.05, **p<0.01, ****p<0.0001. Significance based on ANOVA.
The online version of this article includes the following source data for figure 6:

**Source data 1.** The number of myelinated axons in Veh and Seph groups.

## GADD34 deficiency or Sephin1 treatment does not affect OPC proliferation or microglial recruitment following cuprizone demyelination/remyelination

Recruitment of OPCs is required for spontaneous remyelination, during which OPCs proliferate and migrate to the demyelinated area to mature into functional oligodendrocytes (*Huang and Franklin, 2011*; *Franklin and Ffrench-Constant, 2008a*). Using the OPC marker PDGFRα and the proliferation marker Ki67, we noted the appearance of PDGFRα+ OPCs and proliferating OPCs (PDGFRα+/Ki67 +) near the demyelinated lesions (W5) and remyelinated areas (W8) in the corpus callosum of double-transgenic mice, which was not affected by GADD34 deficiency (*Figure 7A–C*) or Sephin1 treatment (*Figure 7D–F*). Interestingly, at week 5 of cuprizone exposure, the induction of IFN-γ by Dox removal (IFN-γ+) diminished the number of proliferating OPCs (PDGFRα+/Ki67+) in vehicle-treated mice ($p<0.05$), but no difference between vehicle and Sephin1 was found in the number of OPCs in the presence of IFN-γ (*Figure 7D,E*).

Studies have indicated that recruitment of microglia is required for myelin clearance and efficient initiation of remyelination (*Neumann et al., 2009*; *Voss et al., 2012*; *Gudi et al., 2014*). To examine the activation of microglia, we immuno-stained sections of corpus callosum with the microglia marker IBA1 and quantified the density of these cells in the lesions. A significantly stronger microgliosis was observed in the corpus callosum of double-transgenic mice after 5 weeks of cuprizone chow (W5) and activated microglia persisted after 3 weeks of remyelination (W8) (*Figure 7—figure supplement 1* and *Figure 7—figure supplement 2*). Interestingly, in the group of GADD34 KO and of Sephin1 treatment, the number of IBA1+ cells at W8 was significantly increased in the presence of IFN-γ than those in the absence of IFN-γ (*Figure 7—figure supplement 1A,C* and *Figure 7—figure supplement 2A,C*), but we did not detect a statistical change in the density of IBA1+ cells between WT mice and KO or between vehicle and Sephin1 treatment (*Figure 7—figure supplement 1A,C* and *Figure 7—figure supplement 2A,C*).

## Combined treatment of Sephin1 and BZA accelerated remyelination in the presence of IFN-γ

As mentioned above, a number of small molecules have been identified that promote OPC maturation and CNS remyelination as shown in non-inflammatory models of demyelination, and here we have demonstrated that ISR enhancement permits remyelination in the presence of inflammation (*Figures 3–6*). We next tested the hypothesis that the effectiveness of a remyelination-enhancing agent would be augmented in an inflammatory environment when combined with an ISR enhancing agent. BZA is a selective estrogen receptor modulator (SERM) that is currently FDA-approved to be used in combination with conjugated estrogen in menopausal women (*McKeand et al., 2014*). Recently, Rankin et al. showed that BZA significantly enhances OPC differentiation and accelerates remyelination in the lysolecithin non-inflammatory demyelination model (*Rankin et al., 2019*). To test whether the effectiveness of BZA would be enhanced in the presence of inflammation by ISR modulation, we combined BZA and Sephin1 treatments in our cuprizone GFAP-tTA;TRE-IFN-γ double transgenic model. We began the daily administration of BZA and Sephin1 3 weeks after the start of the cuprizone diet and withdrawal of Dox from the animal's water (*Figure 2C*). At 3 weeks after cuprizone withdrawal, we found that Sephin1, BZA, and combined Sephin1/BZA treatment significantly increased ASPA+ oligodendrocyte numbers in the corpus callosum during remyelination in the presence of IFN-γ (W8/ IFN-γ+) (Sephin1: $p<0.05$, $\eta^2 = 0.78$; BZA: $p<0.01$, $\eta^2 = 0.67$; Sephin1/BZA: $p<0.05$, $\eta^2 = 0.77$) (*Figure 8A,C*). Accordingly, compared with vehicle controls, a significant increase in the percentage of myelinated axons was noted in the groups treated with Sephin1, BZA, and combined Sephin1/BZA treatment (Sephin1: $p<0.05$, $\eta^2 = 0.81$; BZA: $p<0.05$, $\eta^2 = 0.77$; Sephin1/BZA: $p<0.01$, $\eta^2 = 0.80$) (*Figure 8B,D*). Nonetheless, no difference between treatment groups was observed either for oligodendrocyte survival or number of remyelinated axons, although the number of remyelinated axons in treatment groups reached pre-lesion levels (*Figure 8B,D*). In addition to measuring the percentage of myelinated axons, we also examined myelin thickness using EM analysis of g-ratios after remyelination and using W0 as the pre-lesion reference point. The vehicle-treated mice (W8/IFN-γ+) demonstrated significantly thinner myelin than those at W0 (IFN-γ-). In contrast, corpora callosa axons in mice receiving the combination treatment of

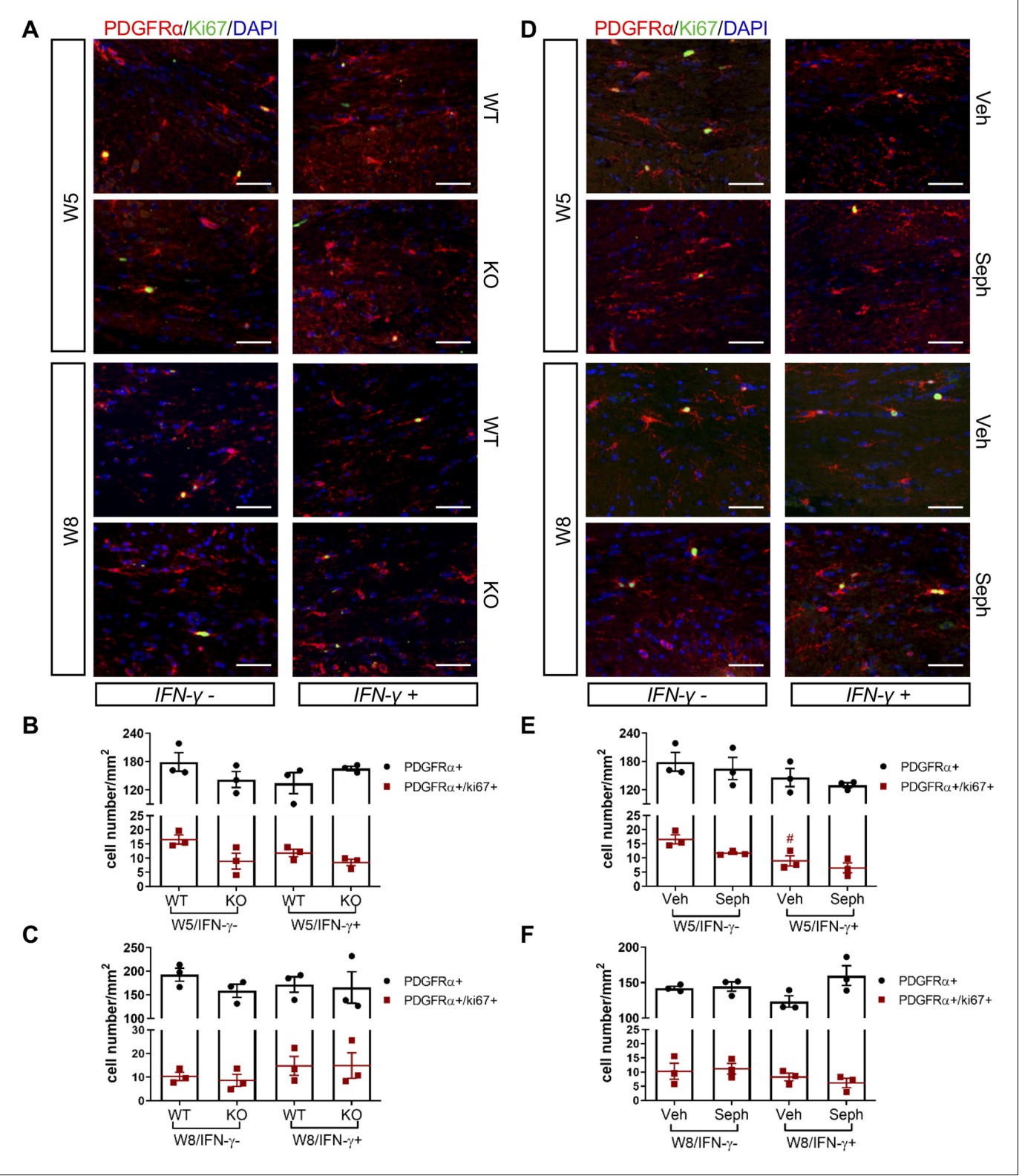

**Figure 7.** GADD34 deficiency or Sephin1 does not affect OPC proliferation during remyelination. (**A**) Immunofluorescent staining for PDGFRα (an OPC marker), Ki67 and DAPI (nuclei) from the corpus callosum of GFAP-tTA;TRE-IFN-γ/GADD34 KO or WT was taken at W5 and W8. Scale bar = 50 μm. Quantification of cells positive for PDGFRα and cells positive for both PDGFRα and Ki67 in the corpus callosum areas of GFAP-tTA;TRE-IFN-γ/GADD34 KO or WT in the absence (IFN-γ-) or presence of IFN-γ (IFN-γ+) at W5 (**B**) and W8 (**C**). Data are presented as the mean ± SEM (n = 3 mice/group). W5 (WT): five males and one female; W5 (KO): four males and two females. W8 (WT): two males and four females; W8 (KO): four males and two females. (**D**)

*Figure 7 continued on next page*

*Figure 7 continued*

Immunofluorescent staining for PDGFRα, Ki67 and DAPI from the corpus callosum of GFAP-tTA;TRE-IFN-γ was taken after either vehicle or Sephin1 treatment at W5 and W8. Quantification of cells positive for PDGFRα and cells positive for both PDGFRα and Ki67 in the corpus callosum areas of GFAP-tTA;TRE-IFN-γ treated with treatment in the absence (IFN-γ-) or presence of IFN-γ (IFN-γ+) at W5 (E) and W8 (F). Data are presented as the mean ± SEM (n = 3 mice/group). W5 (Veh): four males and two females; W5 (Seph): four males and two females. W8 (Veh): five males and one female; W8 (Seph): five males and one female. #p<0.05 (vs. veh from W5/IFN-γ-). Significance based on ANOVA.

The online version of this article includes the following source data and figure supplement(s) for figure 7:

**Source data 1.** The number of total OPCs and proliferating OPCs.
**Figure supplement 1.** GADD34 deficiency does not affect microglial activation during remyelination in the presence of IFN-γ.
**Figure supplement 2.** Sephin1 treatment does not affect microglial activation during remyelination in the presence of IFN-γ.

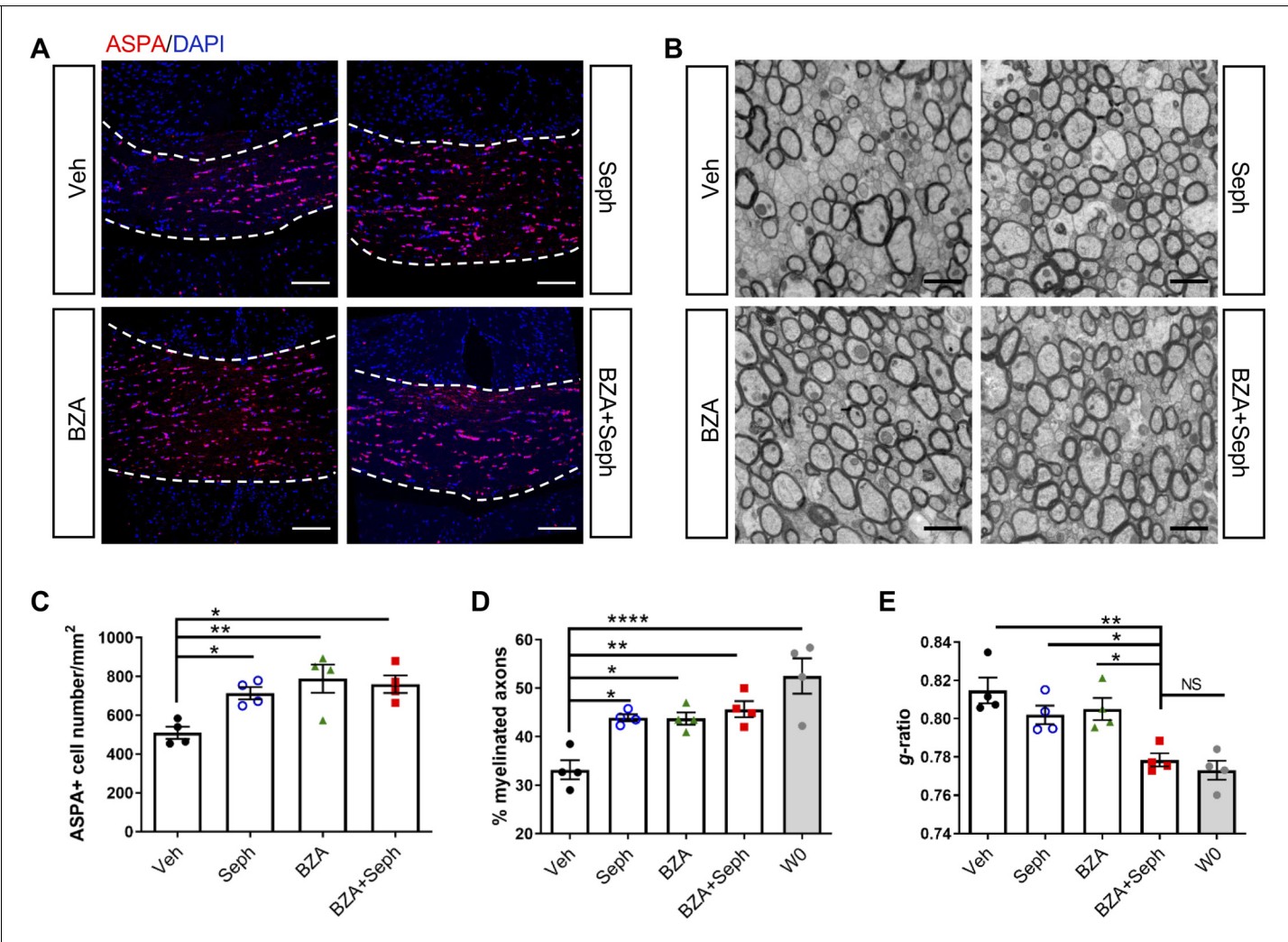

**Figure 8.** Combined treatment of Sephin1 and BZA accelerates remyelination. GFAP-tTA;TRE-IFN-γ mice were released from Dox at W0 and given cuprizone chow for 5 weeks. Treatment with vehicle, Sephin1, BZA or combined BZA and Sephin1 was started at W3, and the corpus callosum was harvested and processed at W0 and after 3 weeks of remyelination (W8). (A) Immunofluorescent staining for ASPA and DAPI in the corpus callosum at W8 in the presence of IFN-γ. Scale bar = 100 μm. (B) Representative EM images of axons in the corpus callosum at W8 in the presence of IFN-γ. Scale bar = 1 μm. Quantifications of the number of cells positive for ASPA (C), percentage of remyelinated axons (D) and g-ratios of axons (E) in the corpus callosum areas at W0 and W8 in the presence of IFN-γ. Two males and two females in each treatment group and four males at W0. Data are presented as the mean ± SEM (n = 4 mice/group). *p<0.05, **p<0.01, ****p<0.0001. Significance based on ANOVA.

The online version of this article includes the following source data for figure 8:

**Source data 1.** The number of ASAP+ cells, myelinated axons and g-ratio of various treatments.

Sephin1 and BZA (g-ratio: 0.778 ± 0.006) reached myelin thickness levels comparable with pre-lesion W0 axons (*Figure 8B,E*). The g-ratios of myelinated axons were significantly lower in the combination treatment group than the vehicle-treated group ($p<0.01$, $\eta^2 = 0.77$) or either of the single treatment group ($p<0.05$, $\eta^2 = 0.72$ vs Sephin1, $p<0.05$, $\eta^2 = 0.72$ vs BZA).

## Discussion

Endogenous remyelination plays a crucial role in preserving axons and restoring neuronal function, but is frequently insufficient in MS. The adverse extracellular environment of MS lesions is thought to contribute to remyelination failure (*Chew et al., 2005*; *Frischer et al., 2015*; *Kuhlmann et al., 2008*; *Franklin and Ffrench-Constant, 2008a*). Efforts to re-populate oligodendrocytes that remyelinate neuronal axons within the CNS will advance reparative therapeutics for MS (*Franklin and Kotter, 2008b*; *Fancy et al., 2010*; *Chari, 2007*). Although many agents identified from high-throughput screening studies appear to accelerate remyelination, it is unclear whether they are capable of initiating repair and restoring neuronal function in a complex inflammatory environment (*Deshmukh et al., 2013*; *Mei et al., 2016*). A recent study, which investigated the differentiation of induced pluripotent stem-cell-derived oligodendrocytes from MS patients, indicated that the inflammation in MS lesions is a major contributor to impaired remyelination, and many remyelinating agents failed to restore impaired differentiation caused by inflammation (*Starost et al., 2020*). Our mouse model used here, which combines primary demyelination induced by cuprizone with CNS delivery of IFN-γ, provides a unique platform to facilitate the assessment of remyelination therapies in the setting of inflammation. In our previous studies, we demonstrated that prolonging the ISR provides protection to mature oligodendrocytes, which maintain the myelin sheath, from inflammatory attack (*Way and Popko, 2016*; *Way et al., 2015*; *Lin et al., 2008*; *Chen et al., 2019*). Herein we investigated whether a similar enhancement of the ISR would provide protection to newly generated remyelinating oligodendrocytes and promote the repair of demyelinated axons in a neuroinflammatory environment.

We have previously shown that a prolonged ISR response protects mature oligodendrocytes during the peak of disease in EAE in a manner not related to direct immunomodulation (*Chen et al., 2019*). Here, we demonstrated that Sephin1 treatment, when initiated at the peak of disease, facilitated the clinical recovery of the EAE animals, which correlated with enhanced remyelination. These results motivated us to examine the remyelination enhancing potential of an augmented ISR in a more controlled model of CNS remyelination.

To further investigate the potential of the ISR to confer protection to remyelinating oligodendrocytes and remyelination during inflammation, we utilized an inducible double-transgenic mouse model (GFAP-tTA;TRE-IFN-γ) to temporally deliver IFN-γ to the CNS in the cuprizone model (*Lin et al., 2006*). The T cell cytokine IFN-γ is a critical inflammatory mediator of MS/EAE pathogenesis (*Lees and Cross, 2007*; *Ottum et al., 2015*). In addition, it has been suggested that IFN-γ could induce immune transitions of OPCs and oligodendrocytes to an inflammatory state with major histocompatibility complex (MHC)-I and MHC-II presentation (*Kirby and Castelo-Branco, 2020*). IFN-γ delivery is required because cuprizone-induced demyelination/remyelination occurs in the absence of an adaptive immune response and thus does not reflect the inflammatory environment in the MS lesions. Importantly, the level of IFN-γ present in the CNS of GFAP-tTA;TRE-IFN-γ double transgenic mice is approximately equivalent to that of EAE mice (data not shown), which suggests that Dox-controlled IFN-γ release is within the range observed in inflammatory demyelination. Our previous studies with GFAP-tTA;TRE-IFN-γ double-transgenic mice demonstrated that ectopic expression of IFN-γ in the CNS results in a reduction in the number of myelinating oligodendrocytes and hypomyelination during development, and results in diminished remyelination following cuprizone-induced oligodendrocyte toxicity (*Lin et al., 2008*; *Lin et al., 2006*). In our current study we found that CNS expression of IFN-γ suppresses the repopulation of oligodendrocytes and results in diminished numbers of remyelinated axons following cuprizone-induced demyelination. In the GFAP-tTA;TRE-IFN-γ model, it takes about 2–3 weeks for IFN-γ levels to become elevated in the CNS following the removal of Dox from the animal's diet (*Figure 2—figure supplement 1*; *Lin et al., 2006*). Since Dox removal occurred at the time of the initiation of cuprizone exposure, the majority of mature oligodendrocytes had already been lost (W3) by the time IFN-γ levels became elevated. Therefore, in this model,

mainly newly generated remyelinating oligodendrocytes are exposed to IFN-γ. Using the same mouse model, our group previously showed that the effect of IFN-γ on the remyelination process in the cuprizone model is associated with an activated ISR response in remyelinating oligodendrocytes (*Lin et al., 2005*; *Lin et al., 2006*). Indeed, GADD34 deficiency or Sephin1 treatment, both of which prolong the ISR, significantly increased the number of remyelinating oligodendrocytes and myelinated axons in the presence of IFN-γ. Nevertheless, the enhancement of the ISR had no effect on remyelination in the absence of IFN-γ. This is consistent with a recent study that showed that guanabenz, a drug closely related to Sephin1 (*Das et al., 2015*), did not improve remyelination in the non-inflammatory cuprizone model of demyelination (*Thompson and Tsirka, 2020*).

The genetic and pharmacologic approaches used here to modulate the ISR do not target specific cell types in the CNS. The increased numbers of remyelinating oligodendrocytes that result from an enhanced ISR might originate from increased OPC survival in the lesion or from more efficient OPC differentiation to mature oligodendrocytes. We observed that proliferating OPC numbers at the peak of cuprizone-induced demyelination were diminished by the presence of IFN-γ, which is consistent with previous findings that showed that IFN-γ predisposed OPCs to apoptosis (*Chew et al., 2005*; *Kirby et al., 2019*). Nevertheless, prolonging the ISR by the *Gadd34* mutation or by Sephin1 treatment did not alter the number of proliferating OPCs in the presence of inflammation. Therefore, it is likely that the benefits of ISR enhancement on remyelination in the presence of inflammation are primarily due to the protection of actively myelinating oligodendrocytes, similar to what we have observed during developmental myelination (*Chew et al., 2005*; *Lin et al., 2005*).

The FDA-approved SERM, BZA, has been shown to be capable of enhancing OPC differentiation and remyelination independently of its estrogen receptor (*Rankin et al., 2019*). We show here that BZA can promote remyelination in the presence of the inflammatory cytokine IFN-γ, which provides valuable preclinical support to the current MS clinical trial of BZA as a remyelination-enhancing agent (ClinicalTrials.gov Identifier: NCT04002934). Encouragingly, we found that the combination treatment of BZA and Sephin1 resulted in more axons with thicker myelin, indicating a more substantial recovery. Although with our current data we cannot exclude the possibility that this combination treatment might protect mature oligodendrocytes and myelin during cuprizone-induced demyelination, we believe that because the treatment was started at the time-point of substantial oligodendrocyte loss and demyelination in the corpus callosum, it is more likely that the primary beneficial effect is on the repopulation of oligodendrocytes and remyelination. The myelin sheaths of remyelinated axons are uniformly thin regardless of axon diameter, but the underlying mechanism that controls the thickness of remyelinated axons is unknown (*Fancy et al., 2011*). Although previous studies have shown that the transgenic overexpression of neuregulin (*Nrg*) or the conditional knockout of *Petn* increases myelin sheath thickness in developmental myelination, the remyelinated myelin sheaths remain thin in these models (*Brinkmann et al., 2008*; *Harrington et al., 2010*). Increased myelin thickness after the combination treatment of BZA and Sephin1 in our inflammatory de/remyelination model suggests that remyelination can be more efficiently induced by using therapies promoting OPC differentiation in combination with therapies that protect the remyelinating oligodendrocytes against inflammatory environment.

Effective MS therapies need to both suppress the aggressive CNS inflammation and promote restoring neuronal functions, including remyelination (*Rodgers et al., 2013*; *Titus et al., 2020*). Our current study demonstrates that the ISR modulator Sephin1 protects remyelinating oligodendrocytes in the presence of an inflammatory environment, leading to remarkably improved remyelination. In our previous EAE study, we showed that Sephin1 protects mature oligodendrocytes, myelin and axons, in addition to indirectly dampening the CNS inflammation that drives EAE (*Chen et al., 2019*). We also showed that combining Sephin1 with the anti-inflammatory MS drug IFN-β resulted in an additive effect in alleviating EAE. Combining current and previous findings, we believe that Sephin1 or similar ISR enhancing strategies will likely provide significant therapeutic benefit to MS patients when combined with current immunosuppressive treatment.

# Materials and methods

## Key resources table

| Reagent type (species) or resource | Designation | Source or reference | Identifiers | Additional information |
|---|---|---|---|---|
| Strain, strain background (*M. musculus*) | C57Bl/6J | Jackson lab | RRID:IMSR_JAX:000664 | |
| Strain, strain background (*M. musculus*) | GFAP-tTA mice | *Lin et al., 2004* | RRID:IMSR_JAX:005964 | |
| Strain, strain background (*M. musculus*) | TRE-IFN-γ mice | *Lin et al., 2004* | RRID:IMSR_JAX:009344 | |
| Strain, strain background (*M. musculus*) | *Ppp1r15a-/-*(GADD34 KO) mice | Gift from David Ron | RRID:MGI:3040935 | |
| Chemical compound, drug | doxycycline | Sigma-Aldrich | Cat # D9891 | |
| Chemical compound, drug | 0.2% cuprizone | Envigo | Cat # TD.160049 | |
| Chemical compound, drug | Sephin1 | Apexbio | Cat # A8708 | |
| Chemical compound, drug | bazedoxifene acetate | Sigma-Aldrich | Cat # PZ0018 | |
| Chemical compound, drug | MOG $_{35-55}$ peptide | Genemed synthesis | Cat # MOG3555-P-5 | |
| Chemical compound, drug | pertussis toxin | List Biological Laboratories | Cat # 179 | |
| Sequence-based reagent | *Gapdh*-f | This paper | qPCR primer | TGTGTCCGTCG TGGATCTGA |
| Sequence-based reagent | *Gapdh*-r | This paper | qPCR primer | TTGCTGTTGAA GTCGCAGGAG |
| Sequence-based reagent | *Ifng*-f | This paper | qPCR primer | GATATCTGGAG GAACTGGCAAAA |
| Sequence-based reagent | *Ifng*-r | This paper | qPCR primer | CTTCAAAGAGTCT GAGGTAGA AAGAGATAAT |
| Antibody | Anti-MBP (mouse monoclonal) | Abcam | Cat # ab24567 RRID:AB_448144 | (1:700) |
| Antibody | Anti-ASPA (rabbit polyclonal) | Genetex | Cat # GTX113389 RRID:AB_2036283 | (1:500) |
| Antibody | Anti-Ki67 (rabbit polyclonal) | Abcam | Cat # AB15580 RRID:AB_443209 | (1:100) |
| Antibody | Anti-PDGFR-alpha (mouse monoclonal) | BD Biosciences | Cat # 558774 RRID:AB_397117 | (1:100) |
| Antibody | Anti-Iba1 (rabbit polyclonal) | Wako Pure Chemical | Cat # 019–19741 RRID:AB_839504 | (1:500) |
| Software, algorithm | ImageJ | National Institutes of Health | RRID:SCR_003070 | |
| Software, algorithm | Prism 6.0 | Graphpad | RRID:SCR_002798 | |

## Animal study

Six-week-old female C57BL/6J mice were purchased from the Jackson Laboratory (Bar Harbor, ME, USA). The mice were allowed to acclimate for 7 days prior to the EAE experiment. The GFAP-tTA mice on the C57BL/6J background were mated with the TRE-IFN-γ mice on the C57BL/6J background to produce GFAP-tTA; TRE-IFN-γ double-transgenic mice (*Lin et al., 2006*; *Lin and Popko, 2009*; *Lin et al., 2004*). Moreover, GFAP-tTA mice or TRE-IFN-γ were crossed with *Ppp1r15a-/-* (GADD34 KO) mice (*Kojima et al., 2003*), and the resulting progeny GFAP-tTA; *Ppp1r15a-/-* were crossed with TRE-IFN-γ; *Ppp1r15a-/-* to obtain double-transgenic mice that were homozygous for the *Ppp1r15a* mutation. To prevent transcriptional activation of IFN-γ, 0.05 mg/ml doxycycline (Dox, Sigma-Aldrich, St. Louis, MO, USA) was added to the drinking water and provided ad libitum from conception. Animals used in this study were housed under pathogen-free conditions at controlled temperatures and relative humidity with a 12/12 hr light/dark cycle and free access to pelleted food and water. All animal experiments were conducted in accordance with ARRIVE guidelines and in complete compliance with Animal Care and Use Committee guidelines of the University of Chicago and Northwestern University.

The mice were randomly assigned to the different experimental groups. Sephin1 (free base) was purchased from Apexbio (Houston, TX, USA), and BZA from Sigma-Aldrich. Stock solutions of Sephin1 (24 mg/ml) and BZA (10 mg/ml) in dimethyl sulfoxide (DMSO) were stored at −20C˙. Final solutions were prepared in sterile 0.9% NaCl (DMSO concentration: 1%) for animal treatment.

## EAE immunization and treatment

EAE was induced in 7-week-old female C57BL/6J mice by subcutaneous flank administration of 200 μg MOG$_{35-55}$ peptide (Genemed synthesis, San Antonio, TX, USA) emulsified with complete Freund's adjuvant (CFA) (MOG$_{35-55}$/CFA) (BD Biosciences, San Jose, CA, USA) supplemented with inactive *Mycobacterium tuberculosis* H37Ra (BD Biosciences). Intraperitoneal (i.p.) injections of 200 ng pertussis toxin (List Biological Laboratories) were given immediately after administration of the MOG emulsion and again 48 hr later. Mice were blindly scored daily for clinical signs of EAE as follows: 0 = healthy, 1 = flaccid tail, 2 = ataxia and/or paresis, 3 = paralysis of hindlimbs and/or paresis of forelimbs, 4 = tetraparalysis, 5 = moribund or death. Mouse groups were randomized during the treatment. Mice were injected intraperitoneally with Sephin1 or the equivalent amount of vehicle (1% DMSO in 0.9% NaCl) daily starting from the peak of disease (score = 3). Lumbar spinal cords were collected at PID30.

## Cuprizone administration

To induce demyelination, GFAP-tTA;TRE-IFN-γ and GFAP-tTA;TRE-IFN-γ;GADD34-/- double transgenic mice (male and female) were fed with a 0.2% cuprizone diet (Envigo, Madison, WI, USA) starting from 6-week-old. Dox water was discontinued at the time of cuprizone treatment (Week 0, W0). Cuprizone feeding lasted 5 weeks and then mice were placed back on normal chow for up to 3 weeks to allow remyelination to occur. Control mice were maintained on Dox water during the entire experiment. 8 mg/kg of Sephin1 (i.p.) or 10 mg/kg of BZA (gavage) was given daily to the GFAP-tTA; TRE-IFN-γ mice, starting from three weeks of cuprizone exposure (W3). The corpus callosum of each mouse was collected at 3 weeks (W3), 5 weeks (W5), or 8 weeks (W8) after cuprizone feeding was initiated. A total of 90 double transgenic mice were used for the study.

## Quantitative real-time reverse transcription PCR

GFAP-tTA;TRE-IFN-γ mice were perfused with ice-cold phosphate-buffered saline (PBS). Total RNA was isolated from the cerebellum using the Aurum Total RNA Fatty and Fibrous Tissue Kit (Bio-Rad, Hercules, CA, USA). Reverse transcription was performed using the iScript cDNA synthesis kit (Bio-Rad). Quantitative real-time reverse transcription PCR was performed on a CFX96 RT-PCR detection system (Bio-Rad) using SYBR Green technology. Results were analyzed and presented as the fold induction relative to the internal control primer for the housekeeping gene *GAPDH*. The primers (5'–3') for mouse gene sequences were as follows: *Gapdh*-f: TGTGTCCGTCGTGGATCTGA, *Gapdh*-r: TTGCTGTTGAAGTCGCAGGAG; *Ifng*-f: GATATCTGGAGGAACTGGCAAAA, *Ifng*-r: CTTCAAAGAG TCTGAGGTAGAAAGAGATAAT.

## Immunostaining

GFAP-tTA; TRE-IFN-γ mice were initially perfused with PBS only before cerebellum harvesting. The same mice were then perfused with 4% paraformaldehyde (Electron Microscopy Sciences, Hatfield, PA, USA) in PBS for 15 min. The brains were removed and cut coronally at approximately 1.3 mm before the bregma. The posterior parts of the brains were post-fixed overnight and embedded in O. C.T. compound (Sakura Finetek, Torrance, CA, USA). The tissue was sectioned in a series of 10 μm on a cryostat. Cryosections were treated with acetone at −20℃, then blocked with PBS containing 5% goat serum and 0.1% Triton X-100, and incubated overnight with the primary antibodies at 4℃. Sections were incubated with secondary antibodies for 1 hr at room temperature. Coronal sections at the fornix region of the corpus callosum corresponding to Sidman sections 241-251 (*Sidman et al., 1972*). Primary antibodies include the following anti-MBP (Abcam, ab24567, 1:700), anti-ASPA (Genetex, GTX113389, 1:500), anti-Ki67 (Abcam, AB15580, 1:100), anti-PDGFR-alpha (BD Biosciences, 558774, 1:100), and anti-Iba1(Wako Pure Chemical, 09–19741, 1:500). The fluorescent stained sections were scanned with Olympus VS-120 slide scanner and quantified by ImageJ. At least three serial sections of corpus callosum were quantified. The representative fluorescent images were acquired under Nikon A1R confocal microscope. The investigators were blinded to allocation of treatment groups in the processes of image capture and the following quantification.

## Electron microscopy (EM)

For GFAP-tTA;TRE-IFN-γ mice, the anterior parts of the brains were immersed into EM buffer for two weeks at 4℃. EM buffer contains 4% paraformaldehyde (Electron Microscopy Sciences), 2.5% glutaraldehyde (Electron Microscopy Sciences) in 0.1 M sodium cacodylate (Electron Microscopy Sciences) at pH 7.3. The sections corresponding to the corpus callosum were trimmed, and postfixed in 1% osmium tetroxide (Electron Microscopy Sciences) in 0.1 M Sodium Cacodylate. Sections were then dehydrated in ethanol, cleared in propylene oxide, and embedded in EMBed 812 resin (Electron Microscopy Sciences). EAE mice were perfused with EM buffer, and then the lumbar spinal cords were processed, embedded, and sectioned as above. Semi-thin sections were stained with toluidine blue. Samples were next ultrathin sectioned on a Leica EM UC7 ultramicrotome. Grids were examined on a FEI Tecnai Spirit G2 transmission electron microscope. We calculated the total percentage of remyelinated axons averaged from 10 images (area = 518.3504 μm$^2$) in each mouse. G-ratio was calculated as the ratio of the inner diameter to the outer diameter of a myelinated axon; a minimum of 300 fibers per mouse was analyzed. The investigators were blinded to allocation of treatment groups in the processes of image capture and the following quantification.

## Statistical analysis

Statistical tests were performed in Prism eight software. No statistical methods were used to predetermine sample size. Each n value represents individual animal. All data were presented as mean ± SEM (standard error of mean). Multiple comparisons were carried out by one-way ANOVA followed by Tukey's post hoc test; single comparisons were evaluated by unpaired t-test. Cumulative scores of EAE mice were analyzed using Kolmogorov-Smirnov method. Differences were considered statistically significant when $p < 0.05$. The effect size was reported as eta squared ($\eta^2$), referring to effect size as small ($\eta^2 = 0.01$), medium ($\eta^2 = 0.06$), and large ($\eta^2 = 0.14$) (*Cohen, 1988*; *Ellis, 2010*).

## Acknowledgements

The authors acknowledge the members of Dr. Raj Awatramani's lab for their assistance with the VS120-S6-W slide loader system and Dr. Hongtao Chen and Mr. Lennell Reynold from the Center for Advanced Microscopy at Northwestern University for their technical support. We thank Erdong Liu for EM tissue sectioning, Dr. Vytas Bindokas from the Integrated Light Microscopy Core Facility, and Yimei Chen from the Advanced Electron Microscopy Core facility at University of Chicago for technical assistance. We also acknowledge Ani Solanki from the Animal Resource Center at University of Chicago for animal study assistance and acknowledge Sharon Way for editing the manuscript. This study was supported by NIH/NINDS R01 NS034939 (BP), the Dr. Miriam and Sheldon G Adelson Medical Research Foundation (JRC and BP) and the Rampy MS Research Foundation (BP).

## Additional information

### Competing interests
Jonah R Chan: has received personal compensation for consulting from Inception Sciences (Inception 5) and Pipeline Therapeutics Inc, and has contributed to and received personal compensation for a US Provisional Patent Application concerning the use of BZA as a remyelination therapy (US Provisional Patent Application Serial Number 62/374,270 (issued 08/12/2016)). Brian Popko: is an inventor on US Patent #10,905,663 entitled "Treatment of Demyelinating Disorders" that describes a small molecule approach to enhancing the ISR as a therapeutic approach for demyelinating disorders. The structure of Sephin1 is included in the molecules covered. The other authors declare that no competing interests exist.

### Funding

| Funder | Grant reference number | Author |
| --- | --- | --- |
| National Institute of Neurological Disorders and Stroke | R01 NS034939 | Brian Popko |
| Dr. Miriam and Sheldon G. Adelson Medical Research Foundation | | Brian Popko Jonah R Chan |
| Rampy MS Research Foundation | | Brian Popko |

The funders had no role in study design, data collection and interpretation, or the decision to submit the work for publication.

### Author contributions
Yanan Chen, Conceptualization, Resources, Data curation, Software, Formal analysis, Supervision, Validation, Investigation, Visualization, Methodology, Writing - original draft, Project administration, Writing - review and editing; Rejani B Kunjamma, Formal analysis, Investigation, Methodology; Molly Weiner, Formal analysis, Investigation; Jonah R Chan, Conceptualization, Writing - review and editing; Brian Popko, Conceptualization, Resources, Supervision, Funding acquisition, Methodology, Project administration, Writing - review and editing

### Author ORCIDs
Yanan Chen https://orcid.org/0000-0001-5510-231X
Jonah R Chan https://orcid.org/0000-0002-2176-1242
Brian Popko https://orcid.org/0000-0001-9948-2553

### Ethics
Animal experimentation: This study was performed in strict accordance with the recommendations in the Guide for the Care and Use of Laboratory Animals of the National Institutes of Health. All of the animals were handled according to approved institutional animal care and use committee (IACUC) protocols (#IS00013825) of Northwestern University. The protocol was approved by the Committee on the Ethics of Animal Experiments of Northwestern University.

### Decision letter and Author response
Decision letter https://doi.org/10.7554/eLife.65469.sa1
Author response https://doi.org/10.7554/eLife.65469.sa2

## Additional files

### Supplementary files
• Transparent reporting form

## Data availability
All data generated or analysed during this study are included in the manuscript and supporting files.

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
