## [Decision Letter]

**Acceptance summary:**

Multiple sclerosis is a demyelinating disease in an inflammatory environment, and understanding the mechanisms that allow oligodendrocyte remyelination in this environment is important for its treatment. This paper demonstrates that the integrated stress response (ISR), a cytoprotective pathway, crucially contributes to the protection of oligodendrocytes and improves remyelination during inflammation. Both pharmacological and genetic approaches were used to prolong the ISR, which led to increased numbers of remyelinating oligodendrocytes and remyelinated axons in animal models of disease.

**Decision letter after peer review:**

Thank you for submitting your article "Prolonging the integrated stress response enhances CNS remyelination in an inflammatory environment" for consideration by *eLife*. Your article has been reviewed by three peer reviewers, and the evaluation has been overseen by a Reviewing Editor and Satyajit Rath as the Senior Editor. The reviewers have remained anonymous.

Essential Revisions:

I am including a summary of the essential revisions required below based on the reviewers' comments.

1) The Introduction requires more clarity with regards to previous literature. As written it is not clear what was known before and how this paper goes beyond the literature.

2) Results and interpretation on the role of sephin1 in EAE is confusing, and needs clarification.

3) Questions about methods and design:

Rationale for specific experimental designs for the cuprizone and combined Sephin1/BZA experiments, ISR indicators in the models, ages and sex of mice used, numbers of mice used, and which steps of EAE and cuprizone experiments were performed blinded.

3) Presentation of figures and question about panels:

For presentation, it was suggested that Figures 2 and 3 be combined. There are also questions about panels in Figures 3C and 5C being duplicated.

4) The combined Sephin1/BZA treatment experiment in Figure 8 need to be clarified both in terms of the rationale and interpretation of the resulting data. All three reviewers point this out. There is a need to clearly describe the rationale for combining BZA and Seph, justifications or revisions of the interpretation of the data and conclusions based on a change in the G-ratio of remyelinated axons without effects on number of ASPA cells and axons and how this can be interpreted as "enhanced" regeneration vs. accelerated regeneration. If it is not sufficiently strong, then the text and Abstract need to be toned down in conclusion.

5) Discussion:

Clearer presentation of conditions during which augmenting ISR is protective of mature and remyelinating oligodendrocytes. Implications of their study for regenerative medicines in the context of drug mediate suppression of inflammation.

---

## [Author Response]

Essential Revisions:I am including a summary of the essential revisions required below based on the reviewers' comments.1) The Introduction requires more clarity with regards to previous literature. As written it is not clear what was known before and how this paper goes beyond the literature.

We thank the reviewers for alerting us to the need to enhance the clarity of the Introduction. We have edited this section in the revised manuscript.

2) Results and interpretation on the role of sephin1 in EAE is confusing, and needs clarification.

We thank the reviewer for this recommendation. We have reworded this section in the revised manuscript (Results).

3) Questions about methods and design:Rationale for specific experimental designs for the cuprizone and combined Sephin1/BZA experiments.

We thank the reviewer for the opportunity to clarify this point. We initiated the combined treatment after three weeks of cuprizone exposure because substantial oligodendrocyte loss and demyelination has occurred in the corpus callosum at this time point. Although we cannot rule out a potential protective effect of the drug treatments on mature oligodendrocytes and myelin, we suspect that because extensive pathology is already present at week 3, the primary protective effect of ISR enhancement in this model is on the repopulation of oligodendrocytes and remyelination in the presence of inflammation. We have revised the manuscript (Discussion).

ISR indicators in the models.

We thank reviewer for this recommendation. We have previously reported that Sephin1 can prolong the phosphorylated status of eIF2α (p-eIF2α) in IFN-γ stressed oligodendrocytes in culture. We also observed a significant increased number of p-eIF2α positive oligodendrocytes in the spinal cords of Sephin1 treated EAE mice, compared to EAE controls (Chen et al., 2019). In addition, using this GFAP/tTA;TRE/IFN-γ double-transgenic transgenic model, our group previously showed that the effect of IFN-γ on the remyelination process in the cuprizone model is associated with an activated ISR response in oligodendrocytes (Lin et al., 2006). In the presence of IFN-γ, the levels of ISR markers (CHOP and p-eIF2α) were significantly increased during remyelination; whereas the ISR does not appear activated in the cuprizone model in the absence of IFN-γ. Furthermore, our previous report demonstrated that the genetic disruption of the ISR inhibited remyelination in the cuprizone model in the presence of IFN-γ (Lin et al., 2006). In the current manuscript we examined whether prolonging the ISR response would enhance remyelination in an inflammatory environment. We have more clearly described this background information in the revised manuscript.

Ages and sex of mice used.

We thank the reviewer for this comment, and we appreciate the opportunity to discuss this issue further. We agree that W8 (14 weeks of age) age-matched mice, in the absence of cuprizone or IFN-γ, would be the ideal control, and we regret not including an examination of such mice for this study. Myelination continues throughout adulthood, such that the 6-week-old timepoint does not precisely reflect the myelin landscape analyzed at the later time points. Therefore, in the revised manuscript we have tried to refer to W0 timepoint (six week of age) more as a reference point, with the primary comparisons being between ISR enhanced (genetic and pharmacological) and non-ISR enhanced animals.

Numbers of mice used.

We thank the reviewers for giving us the opportunity for clarifying these points. We used the same mice for the histology study and EM study; the detailed information can be found in the Materials and methods section. We used the WT mice at W0 mice also for W0 in the pharmacological study. We used WT mice at W5/IFN-γ- also for vehicle controls at W5/IFN-γ-. We included the total number of mice in the revised manuscript.

Which steps of EAE and cuprizone experiments were performed blinded.

We thank the reviewers for the recommendation. We included the information of blinded experiments in the revised manuscript.

4) Presentation of figures and question about panels:For presentation, it was suggested that Figures 2 and 3 be combined. There are also questions about panels in Figures 3C and 5C being duplicated.

We thank the reviewers for this comment. We understand the desire to consolidate the schematic Figure 2 with the following data figure. Nevertheless, we are concerned that this will reduce the clarity of the manuscript. This schematic figure describes not only the IFN-γ inducible system that we used throughout the manuscript (panel A), but it also describes the genetic (panel B) and pharmacological studies (panel C) that were used in the subsequent figures. We hope the reviewers understand our desire to maintain these schematic diagrams as a separate figure to achieve clarity.

We apologize for our error with the incorrectly placed panel in Figure 3C. We have replaced this with the correct panel in the revised manuscript.

5) The combined Sephin1/BZA treatment experiment in Figure 8 need to be clarified both in terms of the rationale and interpretation of the resulting data. All three reviewers point this out. There is a need to clearly describe the rationale for combining BZA and Seph, justifications or revisions of the interpretation of the data and conclusions based on a change in the G-ratio of remyelinated axons without effects on number of ASPA cells and axons and how this can be interpreted as "enhanced" regeneration vs. accelerated regeneration. If it is not sufficiently strong, then the text and Abstract need to be toned down in conclusion.

We thank the reviewers for giving us the opportunity to clarify this point. BZA, which is a selective estrogen receptor modulator, has been shown to significantly enhance OPC differentiation and CNS remyelination. In our study, the enhancement of the ISR, genetically or with Sephin1, provided protection to remyelinating oligodendrocytes in the presence of inflammation. In other words, BZA promotes remyelination and ISR enhancement permits remyelination to occur in the presence of inflammation. We speculated that remyelination-enhancing agents would be even more effective when combined with Sephin1. We have included a more detailed description of the rationale behind these studies in the revised manuscript (Results). We agree that the differences in g-ratio likely reflect an increase in the rate of remyelination in the presence of BZA and Sephin1. We have now discussed this possibility in the revised manuscript. We have not, however, changed the Abstract, since the statement here is factual, not an interpretation of the results: “the combined treatment of Sephin1 with the oligodendrocyte differentiation enhancing reagent bazedoxifene increased myelin thickness of remyelinated axons to pre-lesion levels”.

6) Discussion:Clearer presentation of conditions during which augmenting ISR is protective of mature and remyelinating oligodendrocytes.

We thank the reviewer for the opportunity to clarify this point. It is correct: the ISR does not appear to be activated in the cuprizone model in the absence of inflammation. Oligodendrocyte protection through an enhanced ISR only occurs when the ISR is activated in our model by the presence of IFN-γ. In our previous study, we showed that Sephin1 protected mature oligodendrocytes (i.e. those that are maintaining a myelin sheath) at the peak of EAE disease, where the ISR is activated by CNS inflammation (Chen et al., 2019). In the GFAP/tTA;TRE/IFN-γ transgenic model, it takes about 2 to 3 weeks for IFN-γ levels to become elevated following the removal of dox from the animal’s diet (Figure 2—figure supplement 1) (Lin et al., 2006). In this study, we remove dox at the time of the initiation of cuprizone exposure; therefore the majority of mature oligodendrocyte have already been lost by the time IFN-γ levels increase. We have clarified this in the revised manuscript (Discussion).

Implications of their study for regenerative medicines in the context of drug mediate suppression of inflammation.

We thank reviewer for the recommendation. Our previous study showed that combining Sephin1 with the first-line anti-inflammatory IFN-β exhibited additive benefit in alleviating EAE. We believe that Sephin1 and other ISR enhancing compounds will likely provide significant reparative benefit to MS patients even when combined with current immunosuppressive treatments. We have addressed this point in the revised Discussion.